# The Complement System in Kidney Transplantation

**DOI:** 10.3390/cells12050791

**Published:** 2023-03-02

**Authors:** Donata Santarsiero, Sistiana Aiello

**Affiliations:** Istituto di Ricerche Farmacologiche Mario Negri IRCCS, 24126 Bergamo, Italy

**Keywords:** complement activation, kidney transplantation, ischaemia/reperfusion injury, delayed graft function, alloresponse, antibody-mediated rejection, complement therapeutics

## Abstract

Kidney transplantation is the therapy of choice for patients who suffer from end-stage renal diseases. Despite improvements in surgical techniques and immunosuppressive treatments, long-term graft survival remains a challenge. A large body of evidence documented that the complement cascade, a part of the innate immune system, plays a crucial role in the deleterious inflammatory reactions that occur during the transplantation process, such as brain or cardiac death of the donor and ischaemia/reperfusion injury. In addition, the complement system also modulates the responses of T cells and B cells to alloantigens, thus playing a crucial role in cellular as well as humoral responses to the allograft, which lead to damage to the transplanted kidney. Since several drugs that are capable of inhibiting complement activation at various stages of the complement cascade are emerging and being developed, we will discuss how these novel therapies could have potential applications in ameliorating outcomes in kidney transplantations by preventing the deleterious effects of ischaemia/reperfusion injury, modulating the adaptive immune response, and treating antibody-mediated rejection.

## 1. Introduction

Kidney transplantation is the therapy of choice for patients who suffer from end-stage renal diseases. The process of transplantation involves critical events and inflammatory reactions that occur: (1) before organ procurement, (2) during ex vivo organ preservation, and (3) in kidney allograft recipients. Every step is crucial and can potentially damage the graft.

There is growing evidence that activation of the complement system can occur in each of these stages [1,2,3,4].

In this review, we will first provide an overview of the complement system and the three pathways of activation, highlighting which pathway and which complement components have a detrimental role in each phase of the transplantation process. We will also examine potential triggers of complement activation in renal transplantation and review current and upcoming therapeutic strategies that aim to inhibit the complement system and improve transplant outcomes. 

In doing that, we will also highlight how the blockade of complement activation could be exploited to improve the effectiveness of novel anti-ischemic approaches (e.g., the procedure performed by ex vivo normothermic machine perfusion) or pro-tolerogenic cell therapies (e.g., the post-transplant infusion of MSCs in the case of deceased donor transplantation).

## 2. Overview of the Complement System

The complement system is an essential part of innate immunity. It consists of a family of soluble proteins and membrane-expressed receptors and regulators (as summarised in Figure 1) that are widely distributed and operate in the circulation, in tissues, on cell surfaces, and within cells [5,6].

The complement system, initially considered only a supportive part of immunity, is shown to play important roles in almost every step of immune reactions [7,8]. Today, we recognise over 50 proteins as belonging to the complement system, including plasma components, intracellular proteins, and membrane-bound receptors and regulators, which form a tightly cooperative surveillance network that exerts several functions [9].

Complement is involved in host defence against infection, through: (1) opsonisation of pathogens by C3b, iC3b, C3d, and C4b fragments that are covalently bound to target surfaces to boost phagocytosis [8,10,11,12]; (2) chemotaxis and the activation of leucocytes through the production of potent proinflammatory molecules (the anaphylatoxins C3a and C5a); and (3) direct lysis of bacteria or infected self-cells through the terminal membrane attack complex (MAC, C5b-9) [10]. Secondly, complement can be considered a bridge between innate and adaptive immunity [13]: for example, complement can increase antibody responses and strengthen the immunological memory because C3 receptors are expressed on B cells, antigen-presenting cells (APC), and follicular dendritic cells [14,15,16,17]. Third, complement is essential for the clearance of apoptotic/necrotic, ischaemic, or damaged self-cells (by C1q or C3 binding to cell surfaces) throughout the resolution of the inflammatory reaction, but also during many physiological processes, including development, tissue remodelling, and the maintenance of homeostasis [18,19,20].

In serum and interstitial fluids, complement proteins largely circulate in an inactive form. However, in response to pathogen-associated molecular patterns (PAMPs) or damage-associated molecular patterns (DAMPs), they become activated through a sequential cascade of reactions [7,21]. Indeed, despite the lack of specificity that characterises the innate immune system, complement can selectively recognise pathogens and damaged self-cells through different types of pattern recognition molecules (PRMs), which trigger the initiation of the three pathways of complement activation: the classical (CP), the lectin (LP), and the alternative pathway (AP) [10]. All of these pathways converge in the formation of C3 convertase, independently of the initial danger signal, leading to the production of C3a and C3b. The three pathways of complement activation are schematically overviewed in Figure 1.

The CP initiates when C1q of the C1 complex (which comprises six C1q molecules and two molecules of each of the serine proteases, C1r and C1s) recognises pentraxins, apoptotic, and necrotic cells, or the Fc portion of IgM or IgG antibodies of circulating immune complexes, or of pathogen- or cell-bound immunoglobulins. These interactions lead to the sequential activation of C1r and C1s, which cleave C4 and C2, generating the CP C3 convertase (termed either C4bC2b or C4bC2a in the literature) [22].

The activation of LP relies on the recognition of molecular patterns—such as carbohydrates or other ligands that are expressed on microorganisms but also injured cells [23]—by mannose-binding lectin (MBL), ficolins, or collectins [24]. These PRMs are complexed with mannose-associated serine proteases (MASP-1, MASP-2, and MASP-3), whose activation leads to the cleavage of C4 and C2 to form the LP C3 convertase, in a reaction analogous to CP [10].

Unlike the CP and LP, the AP is continuously and non-specifically activated at low levels in plasma through a process called “tick-over” that allows the system to remain ready for rapid activation when needed [10]. In blood, during “tick-over”, low levels of C3 undergo spontaneous hydrolysis to form C3(H_2_O). This form of C3 is able to bind Factor B (FB), which then becomes a substrate for serine protease Factor D (FD). The cleavage of FB to Bb and Ba by FD results in the formation of the fluid phase AP C3 convertase C3(H_2_O)Bb, which, similarly to the CP and LP C3 convertases, can cleave C3 to C3b and C3a. C3b binds covalently to cell surfaces to form membrane-bound C3 convertase, C3Bb [10]. This process is accelerated and sustained by the presence of properdin, which prolongs the survival of C3b [25]. Under physiological conditions, the deposition of C3b via AP is tightly regulated and not allowed on the surface of self-cells, which are protected by complement regulators. On the other hand, AP activation is permitted on activating surfaces, such as the surfaces of pathogens, which are not protected by complement regulators [10,26]. It should be pointed out that the C3b fragment generated as a consequence of the activation of each of the three pathways can interact with FB and, with the action of FD, can generate the AP C3 convertase C3bBb, which therefore is responsible for an amplification loop of the entire complement system, by increasingly converting a large amount of C3 into its split products, C3a and C3b. Through this powerful positive feedback loop, the AP is often the dominant contributor to the overall complement response, even after CP and LP initiation [27,28], since the C3b produced by the CP and LP provides a platform for new AP C3 convertases.

The addition of a C3b molecule to the C3 convertases generates C5 convertases (C4b2b3b for CP and LP, C3bBb3b for AP), which cleave C5 into C5a and C5b, initiating the terminal pathway. Sequential binding of C5b to the components C6, C7, C8, and multiple copies of C9 molecules forms the membrane attack complex (MAC, C5b-9), which is responsible for the direct lysis of the pathogen or, on target cells, culminates in cell activation or lysis [29].

In addition to the C5b-9 complex, which strikes target cells directly, the other fragments generated in the course of complement activation carry out additional functions upon binding to their receptors on target cells. These essential components of the complement system could be classified into complement receptors (CR1, CR2, CR3, and CR4) and anaphylatoxin receptors (C3aR, C5aR1, and C5aR2) [30]. The CR1 molecule has multiple functions, depending on where it is expressed. For example, it was observed as being able to control the activity of T and B cells, to increase the opsonisation activity of phagocytes, and to promote the clearance of immunocomplexes through binding to C3b and C4b [31]. In addition, CR1 also blocks complement activation in different ways: it destabilises and enhances the decay of the C3 and C5 convertases through binding C3b and C4b, it acts as a cofactor for Factor I-mediated inactivation of C3b, and it may regulate the MBL pathway, acting as an MBL receptor [31]. Together with CR1, CR2 signalling is closely related to the activity of B cells and, in its absence, memory B cell survival is markedly impaired [32]. CR3 and CR4 are often co-expressed on the myeloid subset of leucocytes, but they are also found on natural killer cells and activated T and B cells [33]. They play crucial roles in opsonisation by binding to their ligands (C3b and C3 degradation fragments iC3b, C3dg, and C3d), and in cell adhesion [34].

C3a and C5a receptors are expressed mainly on myeloid cells, but also on endothelial and some parenchymal cells, such as tubular epithelial cells (TECs) [35]. These receptors trigger systemic inflammatory responses, including vascular changes and the chemotaxis of immune cells [36]. The C5a/C5aR1 axis was also shown to be a key player in endothelial thromboresistance loss in several pathological conditions, from genetic rare diseases to viral infections [37].

The powerful effects of complement activation have the harmful potential to also damage the host. Indeed, this defence system may be damaging in certain situations (such as during ischaemia/reperfusion injury) in which complement activation can cause autologous injury. Hence, strict regulation of all pathways and steps in complement activation is necessary to ensure that healthy host cells are spared an aberrant complement-mediated attack. Several complement negative regulators circulate in the blood. C1 inhibitor (C1-inh) and C4b-binding protein (C4BP) control the activation of CP and LP. Factor H (FH) is the primary regulator of the AP, both in the fluid phase and on cell surfaces, and it is a cofactor of Factor I and essential for protecting the host from spontaneous AP activation [24]. Complement negative regulators also include membrane-bound molecules, such as complement receptor 1 (CR1/CD35), decay accelerating factor (DAF/CD55), and membrane cofactor protein (MCP/CD46) [10]. All these regulators mediate the degradation of complement convertases or the inactivation of complement split products, preventing the formation of complement effectors C3a, C3b, C5a, and C5b-9.

A stressful event, such as transplantation, can induce an imbalance in the activation/regulation components of the complement system, leading to a proinflammatory milieu that could eventually lead to graft injury.

## 3. Involvement of the Complement System in Transplantation

During the whole process of transplantation, several events crucially impact graft function and survival and can potentially undermine the overall outcome. Even the steps that must be taken before surgical kidney implantation play a significant role. Indeed, the initial condition of both the donor and the recipient, as well as the organ preservation techniques used, are closely associated with graft quality and outcome. Firstly, the different types of donors, namely living or deceased donors—depending on whether donation occurs after brain death or cardiac death—result in differing organ quality. Regarding kidney transplant recipients, their medical condition and pharmacological treatment until a suitable transplant becomes available also significantly influence graft outcomes. Before surgical implantation, the time and method of graft preservation are crucial and delicate steps in the process of kidney transplantation. Once the organ is implanted in the recipient, the graft finally undergoes reperfusion. From this moment onwards, the graft encounters and inevitably activates the recipient innate and adaptive immune systems, which can potentially induce graft injury and rejection of the donor organ.

An increasing number of studies found that the various components of the complement cascade are crucial players during each step of transplantation [38].

### 3.1. Complement Activation in Donor Kidneys

The clinical condition of the donor and graft have a considerable impact on graft quality and transplant outcomes. In living donation, organs are obtained from healthy people and transplanted following rigorously planned and synchronised surgical procedures. With transplants from deceased donors, which follow brain death (DBD) and cardiac death (DCD), there are frequently pre-existing medical problems, and the organs undergo serious haemodynamic changes, such as prolonged warm ischaemia (in DCD donors) and cell injury with subsequent release of DAMPs and inflammatory responses, which may result in the activation of complement.

Transcriptomic analysis of kidney biopsies revealed activation of the complement cascade in both DBD and in DCD donors before organ retrieval and before the cessation of blood circulation. Kidneys from healthy donors did not exhibit such complement activation [39]. Early studies demonstrated the presence of complement C3 in kidneys from DBD rats [40]. C3d deposition was detected in renal biopsies from human DBD donors before reperfusion, suggesting that C3d was deposited as a result of brain death [41]. Moreover, complement activation via AP was observed in sera from deceased donors compared to sera from healthy subjects [42].

Most studies about complement activation following brain death focused on C3, but there is emerging evidence that downstream activation products are equally important. Significantly higher plasma levels of both C5a and sC5b-9 were reported in DBD donors compared with living donors [43,44] and plasma levels of sC5b-9 in DBD donors were associated with a higher risk of rejection after renal transplantation [43]. The inflammation initiated by brain death is accompanied by the release of C5a in the circulation, increased expression of C5aR in renal tubular cells, and a local inflammatory response [45]. After transplantation, increased levels of circulating sC5b-9 were observed only in recipients of kidney transplants from deceased donors, but not in patients who received kidneys from living donors [46].

In addition, studies showed that the perioperative levels of sC5b-9 could be used as a clinical biomarker of delayed graft function (DGF). Indeed, compared to recipients with early recovery of graft function, recipients with DGF exhibited higher plasmatic levels of sC5b-9 [47].

Notably, in a rat model of brain death, treating DBD donors with soluble complement receptor 1 (sCR1, a complement-inhibiting drug that prevents complement activation at the levels of C3 and C5 convertases) improved graft function after transplantation [48]. A similar result was obtained by treating DBD rat donors with a high dose of C1-inhibitor, which blocks the activation of the CP and LP [49]. Consistent with the above-mentioned studies, a recent study in non-human primates showed that pre-treatment of DBD donors with C1-inhibitor improved kidney allograft functions and prevented the development of DGF [50]. Another preclinical study that used a rat model of brain death showed that pre-treating donors with a monoclonal antibody against Factor B, a key element of the AP, preserved renal function and reduced organ damage and inflammation of the kidney grafts [51]. All these studies suggest that inhibition of the complement system might be an attractive approach to mitigate inflammation and injury in kidneys from deceased donors.

### 3.2. Complement Activation in Transplant Candidates

Complement activation before transplantation may also affect the recipient. Indeed, abnormal complement activity in kidney transplant candidates may be the result of complement-driven kidney diseases (such as atypical haemolytic uremic syndrome (aHUS), C3 glomerulopathy, anti-neutrophil cytoplasmic antibodies (ANCA)-associated glomerulonephritis) [52], and other conditions, including diabetes [53].

Moreover, until a kidney transplant becomes available, many patients are on maintenance haemodialysis. Complement activation can occur during each haemodialysis session. Indeed, upon coming into contact with blood, the surfaces of the biomaterial of the tubing sets and of the dialysis filters are immediately covered with a layer of plasma proteins through passive adsorption. Negatively charged surfaces tend to bind C1q and properdin, which initiate or modulate complement activation via the CP and AP, respectively [54]. C1q binds to immunoglobulins IgG, which are adsorbed by the membrane dialyser and can activate the complement response by the CP [55].

### 3.3. Ischaemia/Reperfusion Injury (IRI) and Complement Activation

Transplantation is described as “a race against the clock” because of ischaemia [56].

Ischaemia is one of the most common causes of complement activation in kidney transplantation and—combined with reperfusion, which triggers the production of reactive oxygen species [57]—is a major cause of inflammation, graft damage, and DGF.

Organs may undergo ischaemic events in the donor, during organ procurement, transportation and preservation (unless some form of ex vivo organ perfusion is performed), as well as during implantation, until reperfusion is achieved in the recipient [56]. During ischaemia, the kidneys undergo oxygen deprivation and shift to anaerobic metabolism. The resulting acidic conditions interfere with complement regulation, enhancing AP activation [58]. Ischaemia also provokes damage to tubular, endothelial and perivascular cells, with a subsequent massive release of DAMPs, such as hyaluronic acid, fibronectin, and DNA, which can be detected and bound by C1q, MBL, collectins, ficolins, and C3b, thus promoting the activation of the complement system [10,21].

A recent technique that aims to preserve/improve graft quality is based on the use of ex vivo normothermic machine perfusion (NMP) while the organ is waiting to be transplanted. During NMP, the kidney graft is perfused with an oxygenated solution at 37°C. Based on the hypothesis that the complement cascade could be activated through contact with foreign surfaces, which is the case for NMP components, a very recent study evaluated complement activation in discarded porcine and human kidneys subjected to NMP [59]. The authors showed that kidney NMP induced complement activation, as documented by the presence of high levels of C3a, C3d, and sC5b-9 in perfusate samples taken during NMP of both porcine and human kidneys. After 4 h NMP, porcine kidneys with high amounts of sC5b-9 in the perfusate exhibited significantly lower renal graft function and higher levels of tissue damage. Notably, at the end of NMP, C3d deposition was observed in all human renal biopsies, with wider C3d deposits in renal biopsies derived from DBD donors than in those from DCD donors. With the limitation that only a small number of human kidneys were included in the study, Jager et al. showed that complement is clearly activated during NMP and suggested that the inhibition of complement during NMP could be a promising strategy for reducing renal injury and ameliorating graft function before transplantation [59].

The key contribution of the complement system in IRI is widely studied and well documented in animal models. Some studies showed that administering siRNA against C3 protects the kidney against IRI, thus improving renal function [60,61]. While most of the circulating complement components are produced by hepatic synthesis, smaller amounts are generated at extrahepatic sites, such as the kidney tubular epithelium. The local synthesis of C3, the cross point of the three complement pathways, is essential for complement-mediated reperfusion damage, whereas circulating C3 had a negligible effect [62]. This was demonstrated by the fact that kidney isografts from C3-positive donor kidneys transplanted into C3-negative recipients developed widespread tissue damage and severe acute renal failure, whereas C3-deficient mice exhibited only moderate reperfusion damage when transplanted into wild type recipient mice [62].

Sieve et al. showed that, as a consequence of the ischaemia-induced breakdown of glycocalyx, endothelial cells lose regulators of the complement system [63], so they are no longer protected from the injury caused by abnormal complement activation. In addition, evidence that treatment with an anti-C5 antibody preserves the integrity of the glycocalyx and protects kidney function in a mouse model of IRI suggest that a vicious cycle involving IRI, terminal complement activation, and glycocalyx loss may occur [64].

#### 3.3.1. Ischaemia/Reperfusion Injury: The Contribution of the Different Pathways of Complement Activation

One important question is which of the three complement pathways is predominant in the C3 cleavage that occurs in grafts injured by IR.

The AP is generally described as playing a crucial role in renal IRI, as demonstrated by the finding that Factor B (FB)-deficient mice [65] and wild type (WT) mice treated with an anti-FB antibody [66], exhibited less kidney reperfusion damage after warm ischaemia.

Notably, a recent preclinical study documented that defective activation of AP in recipients of an ischaemic renal allograft protects the kidney from IRI. In a mouse model of kidney allotransplantation, Casiraghi et al. showed prolonged kidney allograft survival and better graft function in FB-deficient C57BL/6 mice that received an ischaemic Balb/c kidney allograft [67]. Similar results were obtained when WT recipient mice were treated with an anti-FB antibody. Compared to untreated WT mice, FB-deficient mice exhibited reduced intragraft complement deposition, a lower number of graft-infiltrating neutrophils, and less inflammation and tubular damage [67]. In addition, the results (FB-deficient mice exhibited a much lower degree of intragraft T cell infiltration and lower frequency of donor-specific interferon gamma (IFN-γ)-producing T cells after ex vivo MLR) suggested that a defective AP in recipient mice led to an impaired adaptive immune response against ischaemic allografts.

Factor H (FH), the negative AP regulator, plays an important role in limiting injury in the kidney after IR. Indeed, Goetz et al. found that mice with a heterozygous deficiency for FH exhibited C3 deposition in the tubulointerstitial area and developed more severe acute kidney injury (AKI) than WT mice after IRI [68].

However, it should be kept in mind that the AP may also play a protective role in kidney IRI. Indeed, the lack of properdin (an AP positive regulator) is shown to have detrimental effects on kidneys during the reparative phase that occurs after IRI. Properdin, produced locally by TECs, has crucial functions in the opsonisation of damaged cells and in the regulation of the phagocytic ability of TECs to effectively clear apoptotic cells and reduce inflammation [69].

Conversely, results that show that mice deficient in C4 are not protected against IRI suggest that the CP and/or the LP do not play a major role in kidney IRI [70]. Indeed, C4 has critical functions in both CP and LP through the activation induced by the serine protease C1s or the MASP-2, respectively [71]. Intriguingly, several studies that used animal models provided evidence of the involvement of the LP in kidney IRI. Indeed, DAMPs released by the ischaemic injured tissue are among the ideal activators of the LP [72]. Mice deficient in MBL had dramatically less tissue injury after renal IR compared to WT mice [73]. Moreover, in the transplantation of kidneys from deceased donors, higher pretransplant MBL levels were associated with poorer graft survival [74].

Interestingly, in an isogenic mouse model of renal transplantation, Asgari et al. revealed that C4-bypass MASP-2-dependent LP activation in renal IRI plays a role. Specifically, they found that WT grafts transplanted into MASP-2-deficient mice exhibited better renal function than the WT to WT combination. This phenotype was maintained when WT grafts were transplanted into mice deficient in both MASP-2 and C4, highlighting the involvement of a MASP-2-dependent activation of the LP independently of C4 [75]. These results offer an explanation for the above-mentioned absence of protection from IRI in C4-deficient mice, suggesting that distressing events can occur through MASP-2-dependent activation, which then possibly acts by directly cleaving C3 [76].

Further support for the critical role of LP in renal IRI comes from a novel LP initiator, collectin-11 (CL-11), which is normally present in renal tissue and whose expression is increased after IRI. CL-11 can bind several molecules whose expression is increased in ischaemic tissues. Such CL-11 ligands include both extracellular DNA, which is associated with apoptotic cells [77], and fucosylated molecules that are exposed on the surface of hypoxia-stressed cells of the proximal tubules. CL-11 binding to fucosylated molecules can initiate the LP through the activity of MASP-2 [21,78]. Notably, in a mouse model of renal ischaemia, i.p. administration of a high dose of L-fucose protects the kidney against acute ischaemic injury by preventing CL-11 binding to proximal tubular cells, and thus reducing complement activation [79,80]. Specifically, compared to the irrelevant sugar D-galactose, treatment with L-fucose significantly reduced tubular C3d deposition, tubular necrosis, and the infiltration of inflammatory cells, ameliorating renal function [79].

Overall, AP and LP are the leading routes involved in aberrant complement activation during kidney IRI. More specifically, LP is the main trigger of complement activation, which is then amplified through the AP [80]. LP activation is therefore an attractive target during the immediate post-transplant stage for preventing IRI, with MASP-2 being the best candidate to block, because of the absolute requirement for LP activation in renal IRI [81].

Finally, the activation of either the AP or the LP leads to the assembly of MAC on tubular and endothelial cells, resulting in direct cell destruction and further amplification of inflammation [82]. This suggests that blocking the terminal pathway could be a useful therapeutic intervention against IRI. In this context, there were encouraging results from a rat model of syngeneic kidney transplantation that show that supplementation of the allograft preservation solution with an anti-rat C5 mAb during cold ischaemia time prevents ischaemic injury and the consequent DGF [83]. Specifically, donor kidneys were treated pretransplant with either the anti-rat C5 monoclonal antibody 18A10, or the chimeric molecule CR2-FH (TT30), which is an AP inhibitor. Compared to untreated controls, graft survival improved with both treatments, with better outcomes obtained in the 18A10-treated grafts [83].

A new anti-C5 neutralising antibody coupled with a cyclic RGD peptide was recently developed (Ergidina) [84]. This antibody has a distinctive homing property for endothelial cells. Notably, pre-treatment of rat kidneys with Ergidina was found to be effective in limiting tissue damage following IR [84]. The authors also documented the ex vivo binding of Ergidina to surgically removed kidneys exposed to cold ischaemia, supporting the therapeutic use of Ergidina to prevent post-transplant IRI.

#### 3.3.2. Ischaemia/Reperfusion Injury: The Contributions of Other Components of the Complement System, Regulators and Receptors

Many studies that investigated the involvement of the complement system in IRI focused on the role of complement regulatory proteins expressed on the cell surface. Intact healthy cells express a number of membrane-bound complement regulatory proteins to prevent complement-mediated tissue damage. These include CD55, which accelerates the decay of the C3 and C5 convertases, and CD59, which prevents the assembly of the MAC [85]. Under pathophysiological circumstances, such as during IRI, these control mechanisms are crucial and may be overwhelmed and rendered unable to restrain complement-mediated insults [86].

In a murine model of IRI, it was shown that a genetic deficiency of CD55 and CD59 creates increased susceptibility to acute kidney injury. Indeed, CD55- and CD59-deficient mice exhibited augmented renal IRI, as indicated by elevated blood urea nitrogen levels, histological scores, and neutrophil infiltration [87]. On the other hand, the overexpression of human CD55 and CD59 proteins in mice, or treatment with soluble recombinant human CD55, reduced renal C3 and C9 deposition and conferred protection against IR-induced tissue damage [88].

The complement cascade generates the anaphylatoxins C3a and C5a. These small fragments participate in the pathogenesis of renal IRI by binding to their receptors, C3aR, and C5aR, respectively, and promoting the local production of cytokines and chemokines by both renal tubular epithelial cells and infiltrating immune cells [89,90]. It was found that mice that lack either or both receptors are protected against ischaemic injury. Notably, the best protection was conferred by a deficiency for either C3aR/C5aR1 or C5aR1, suggesting that the terminal product of complement activation C5a plays a prominent role [91]. Indeed, blocking C5aR1 signalling, either by using a siRNA silencing approach or a receptor-specific antagonist, reduced renal function impairment and decreased tissue inflammation in models of warm IRI in rodents [92,93,94]. In addition, in a mouse model of syngeneic kidney transplantation, supplementation of the allograft preservation solution with a C5aR1 inhibitor during cold ischaemia time reduced kidney injury and tubular cell apoptosis [95].

### 3.4. Complement Activation Modulates the Adaptive Immune Response against the Graft

Once in the recipient, the graft is attacked by the host adaptive immune system, which, if left uncontrolled, leads to acute transplant rejection. Several studies highlighted how the complement system and adaptive immunity are intimately interconnected, including in the context of transplantation. Indeed, the complement system can play a role in modulating both the cellular and the humoral immune response against the graft.

#### 3.4.1. The Role of Complement in Regulating T Cell Responses

Through its ability to influence T cell functions, both directly and indirectly, by modulating the activities of antigen-presenting cells (APCs), complement acts as a bridge between the innate and adaptive immune response.

It was shown that DCs are able to synthesise C3 and that C3 synthesis is required for the DC’s capability to stimulate the alloreactive T cell response in vitro and in vivo [96]. Indeed, C3 deficiency in donor DCs favours the polarisation of CD4^+^ T cells toward a Th2 phenotype and leads to delayed rejection of the allograft in a mouse model of skin transplantation [96]. In addition, in a mouse model of kidney transplantation, it was reported that the absence of local synthesis of C3 (achieved by using grafts from C3-deficient donors) determines defective T cell priming and a subsequent weakened immune response against donor antigens, thus resulting in prolonged graft survival [97].

In vitro experiments showed that DCs from C3, Factor B, or C3aR knockout mice, or DCs exposed to a C3aR antagonist, fail to upregulate the major histocompatibility complex (MHC) II, thus establishing defective T cell priming and generating a poor alloreactive CD4^+^ T cell response [98]. In contrast, in the same study, C4-deficient DCs appeared to function normally in stimulating the alloreactive T cell response [98], indicating that the AP plays a major role in guiding DC immunostimulatory capability.

Heeger et al. found that deficiency of the complement inhibitory protein DAF (CD55) on either APCs or on T cells leads to unrestrained C3 and C5 convertase formation and to increased production of the local anaphylatoxins C3a and C5a, with a subsequent higher proliferation of effector T cells [99]. Another study demonstrated that autocrine C3aR1/C5aR1 signalling in DCs is a requisite for TLR-mediated DC maturation, which is required for the induction and amplification of T cell responses [100]. Notably, anaphylatoxin receptors, which are expressed on both T cells and APCs, play a role in T cell expansion [101] and in the release by APCs of interleukin (IL)-12, which is necessary for Th1 activation [102]. Accordingly, studies in mice indicated that a deficiency of C5aR in both donors and recipients [103], or inhibition of C5aR by an antagonist [104], reduced allospecific T cell responses and prolonged renal allograft survival.

More recent studies showed that complement activity is not exclusively confined to the extracellular space. Intracellular complement activity is shown to be involved in the regulation of basic cellular metabolic processes [105]. In T cells, these include homeostasis and differentiation [106].

In resting T cells, Liszewski et al. documented that the protease cathepsin L (CTSL) continuously processes intracellular C3 into bioactive C3a and C3b. When intracellular C3 activation is abrogated by CTSL inhibition, T cells succumb to apoptosis [106]. Following the activation of the T cell receptor and the co-stimulatory CD28 receptor, intracellular C3a and C3b rapidly translocate to the cell surface, where they can bind to C3aR and CD46, respectively [106]. In activated human T cells, autocrine stimulation of complement receptor CD46, and specifically of its intracellular domain CYT-1, by intracellular cleaved C3b, was required for the induction of the amino acid transporter LAT1 and enhanced the expression of the glucose transporter GLUT1, resulting in a metabolic switch toward glycolysis and the activation of the checkpoint kinase mTORC1, both of which are necessary for a Th1 response [107]. In addition, the CD46 transduction signal delivers co-stimulatory signals for optimal cytolytic T cell activity by augmenting the nutrient influx and fatty acid synthesis in CD8^+^ T cells [108].

Importantly, in vitro studies showed that the intracellular C5a-C5aR1 signalling in T cells is involved in the Th1 response [109]. Specifically, it was shown that intracellular C5 activation contributes to the induction of IFN-γ production in CD4^+^ T cells via intracellular C5aR1 engagement, whereas C5aR2 acts as a negative regulator of this process. Intracellularly, C5 activation and the stimulation of C5aR1 are required for the NLRP3 inflammasome assembly that initiates caspase-1-dependent IL-1β secretion, promoting IFN-γ production and Th1 differentiation in an autocrine manner. Thus, T cells incorporated the evolutionarily older complement system and the NLRP3 inflammasome as part of their activation machinery [110]. This underlines the need to test the efficacy of blocking both complement and/or inflammasome as potential approaches to restrain pathogenic and aberrant T cell immune responses, including those directed towards transplanted organs.

Components of the complement system are also involved in changes in T cell activity that sustain the resolution phase of immune responses. Through in vitro studies with healthy human CD4^+^ lymphocytes, Kemper et al. showed that co-engagement of CD46 and CD3 in the presence of IL-2 generated CD4^+^ T cell clones with a T-regulatory 1 (type 1 Treg or Tr1) phenotype [111], defined by the production of IL-10 [112] and suppression of T helper cells [113]. A further study confirmed that CD46 plays a role as a key modulator of adaptive immunity through its ability to provide termination signals, such as IL-10, required for the transition toward the contraction phase of the Th1 response [114].

Other complement components also have the ability to quiet the adaptive immune response, e.g., C1q and the complement immunoglobulin receptor (CRIg).

C1q is present in the serum and is also secreted or expressed on the cell surface of APCs, such as monocytes, macrophages, and monocyte-derived DCs [115,116]. In addition to its traditional defensive roles against infections through the activation of the CP, C1q can also maintain tolerance by regulating adaptive immunity [117]. In studies that compared the whole proteomes in human monocyte-derived DCs from patients enrolled in a clinical trial that evaluated the efficacy of allergen-specific immunotherapy, C1q was identified as a hallmark of regulatory DCs and as a candidate biomarker associated with clinical tolerance induced by allergen immunotherapy [118]. In a further study, the authors showed that C1q treatment in mice with induced allergic asthma did not induce Treg cells. Instead, C1q treatment induced murine plasmacytoid DCs to secrete lower levels of proinflammatory cytokines [119].

CRIg is mainly expressed on macrophages and is recognised not only as a phagocytosis-promoting complement receptor, but also as a T cell inhibitor with anti-inflammatory properties [120]. Indeed, it was suggested that CRIg-mediated signalling is a mechanism through which resident macrophages modulate tissue homeostasis and inflammatory processes by regulating the T cell response [121].

#### 3.4.2. The Role of Complement in Antibody-Mediated Rejection

Humoral rejection is a major/the main cause of long-term kidney graft loss [122,123], and complement can help antibody-mediated rejection on several levels.

ABO and HLA system incompatibilities are the main source of immunological risk in allogeneic transplantation. Natural antibodies that react to ABO blood group antigens are involved in humoral immunity to allografts. These pre-existing antibodies, which are produced without previous exposure to the cognate antigen in a T cell-independent manner [38], are primarily IgM that are strong activators of complement [124]. In addition to natural antibodies, the development of anti-HLA antibodies in transplant recipients can occur in a T cell-dependent manner in HLA-mismatched transplant patients who are already immunised, such as multiparous women or patients with a long history of transfusion [125]. Recipients with pre-formed anti-HLA antibodies have an increased risk of hyperacute or acute antibody-mediated rejection (ABMR) and graft loss [126,127].

Preformed or post-transplant de-novo donor-specific antibodies (DSAs) bind to donor MHC epitopes [30], thus initiating the CP through C1q engagement and then resulting in acute or hyperacute ABMR, with damage to the transplant and potential graft loss. Endothelial cells are the main target of DSAs. Indeed, linear C4d deposition in peritubular capillaries or medullary vasa recta is one of the criteria of the Banff classification score for defining active or chronic ABMR [128]. The cascade activated by complement-fixing antibodies culminates in MAC formation that can cause endothelial cell lysis and subsequent direct damage to the vasculature within the graft [30]. MAC also leads to the upregulation and exposure of P-selectin, tissue factor [129], and von Willebrand Factor on endothelial cells [130], promoting a thrombogenic response. Moreover, some studies show that the insertion of the MAC, rather than inducing cytolysis, can upregulate inflammatory genes, enhancing the capacity of endothelial cells to recruit and activate allogeneic IFN-γ-producing CD4+ T cells, further amplifying the humoral response [131].

Complement also plays a role in supporting B cell maturation and activation. Indeed, binding of C3-opsonised antigens to CR2 expressed on follicular DCs leads to a more efficient retention of antigens in the B cell areas of the germinal centres, which is essential for the generation of a robust secondary antibody response [132]. Moreover, the interaction between CR2 expressed on B cells and C3d-opsonised antigens reduces the threshold for B cell activation and promotes antibody production [133,134].

Notably, in an HLA-sensitised model of kidney transplantation in non-human primates, it was recently shown that transient peri-transplant treatment with a C3 inhibitory peptide (Cp40) significantly improved renal function, prolonged graft survival, and alleviated the detrimental action of complement-fixing DSAs in ABMR [135]. In addition, the C3 blockade modulated T and B cell activation and proliferation, suggesting that there is also an immunomodulatory effect [135,136].

## 4. Complement as a Therapeutic Target in Kidney Transplantation

As it was described here and in other reviews, studies using animal transplant models showed that complement plays a crucial role in IRI, graft dysfunction, and cell- and antibody-mediated transplant rejection [137,138]. These preclinical studies also suggested that inhibition of complement activation could be helpful in each step of the transplantation process, i.e., before organ procurement, during ex vivo organ preservation, and after transplantation in allograft recipients, providing the basis for evaluating complement inhibitors in clinical trials (Figure 2).

Among the complement inhibitors that are under clinical evaluation in different clinical settings, eculizumab and C1 esterase inhibitor (C1-inh) are the most extensively tested.

Eculizumab is a recombinant humanised hybrid IgG2/IgG4 monoclonal antibody that targets C5. The binding of eculizumab to C5 prevents its cleavage, therefore inhibiting both the production of C5a and the assembly of MAC [139]. Eculizumab was approved for treating patients with paroxysmal nocturnal haemoglobinuria (PNH) [140] and atypical haemolytic uremic syndrome (aHUS) [141,142,143], diseases characterised by abnormal and unrestrained complement activation. Studies showed that patients with aHUS who were waitlisted for kidney transplant and received prophylactic eculizumab treatment underwent successful renal transplantation and experienced a reduced incidence of post-transplant dialysis [144,145]. Eculizumab treatment was also helpful in achieving successful kidney transplantation in patients with antiphospholipid antibody syndrome [146,147] or C3 glomerulopathy [148].

Notably, several clinical trials evaluated the efficacy of eculizumab in avoiding and treating ABMR or in preventing DGF in kidney transplantation [149,150,151,152,153,154,155,156,157,158,159,160,161,162]. As reviewed by Qi and Qin, while some trials showed encouraging outcomes, others, such as *NCT01106027*, failed to demonstrate the superiority of eculizumab compared to standard-of-care treatments, such as intravenous immunoglobulin administration and plasmapheresis. These controversies might be due to insufficient eculizumab exposure or to the small cohorts of patients included in the studies [30]. Moreover, results from another clinical trial recently reported that eculizumab administration was safe but not useful in reducing the rate of DGF and in improving early and late post-transplant renal functions [163,164].

Another complement inhibitor that is widely tested in kidney transplantation is C1-inh. This endogenous complement regulatory protein inactivates the serine proteases that operate at the beginning of CP (C1r, C1s) and LP (MASPs). Therefore, unlike eculizumab, which impedes C5 cleavage, C1-inh acts at the first step of the complement cascade, preventing the production of C3 and C4 split products [165]. A double-blinded, randomised clinical trial investigated the safety and efficacy of C1-inh in reducing DGF in patients who receive a kidney transplant from deceased donors. Perioperative treatment for recipients with C1-inh showed promising results, since it was associated with a lower incidence of graft failure and with an improvement in the estimated glomerular filtration rate (eGFR), even 3 years after transplantation [166]. Clinical trials are planned to test the efficacy of C1-inh in transplant patients who are at high risk of DGF. C1-inh will be given perioperatively to adult subjects who receive kidney allografts from deceased donors [167,168] or to DBD donors as a pre-explant treatment [169]. In addition, as reviewed by Berger et al. [170], clinical studies in kidney transplant recipients suggested that C1-inh might have therapeutic value in preventing and treating ABMR [171,172,173,174,175], including in patients who are refractory to other treatments [176,177].

More recent studies include trials with Sutimlimab and Cp40. Sutimlimab (Enjaymo, Sanofi) is an anti-C1 monoclonal antibody that was recently evaluated in a phase I clinical trial. Ten kidney transplant recipients with evidence of late active ABMR and CP activation (C4d deposition, complement-fixing DSA) received four weekly doses of Sutimlimab. Follow-up biopsies performed 32 days after the first infusion showed that the treatment caused a significant decrease in C4d deposition. However, no overall improvement in renal function was detected [178,179]. Cp40, a C3 inhibitory peptide that was found to significantly improve renal function and to prolong graft survival in a non-human primate model of kidney transplantation [135], was evaluated in a phase I trial in healthy volunteers [180]. The results are not yet available, but ABO-incompatible kidney transplantation is mentioned [180] among its potential clinical applications.

To attenuate ABMR in transplanted patients with high levels of preformed anti-HLA antibodies, a distinct strategy based on the use of IdeS is worth mentioning. IdeS, a recombinant *Streptococcus pyogenes*-derived endopeptidase, cleaves all IgG subclasses in the central region. The first step results in the single cleavage of the IgG molecule, in which one heavy chain remains intact. The second step generates a fully cleaved product that cannot mediate complement-dependent cytotoxicity or antibody-dependent cell-mediated cytotoxicity [181]. Promising results from a combined phase I–II clinical trial report that IdeS reduced or eliminated DSAs and permitted HLA-incompatible transplantation in 24 of the 25 highly sensitised patients recruited [181,182,183,184].

More recently, the clinical trial EMPIRIKAL tested the ability of the complement inhibitor Mirococept to reduce the incidence of DGF in recipients of a kidney transplant from deceased donors. Mirococept, a molecule derived from human CR1, offers the advantage of blocking all three complement pathways. In rat models of kidney transplantation, Mirococept was found to inhibit complement-mediated tissue injury and allograft rejection of kidney grafts subjected to either short (30 min) [185] or prolonged (16 h) cold ischaemia time [186]. In the EMPIRIKAL clinical trial, the inhibitor was given ex vivo during allograft preservation [187]. The study was stopped because no significant difference was detected between Mirococept and the control group. However, a subsequent dose saturation study in a porcine model showed that the Mirococept exposure in the EMPIRIKAL trial was insufficient and an optimal dosage was determined, paving the way for further clinical investigation [188].

## 5. Blocking Complement to Help Pro-Tolerogenic Cell Therapies

Novel approaches that aim to modulate anti-donor immune responses and induce transplant tolerance are based on the use of cellular therapies. Among the cellular therapies studied, one based on mesenchymal stromal cells (MSCs) recently emerged as very promising.

In human recipients of kidney transplants from living donors, autologous bone marrow-derived MSC infusion was found to be a safe and practicable strategy to promote a pro-tolerogenic environment [189,190,191] and to prevent acute rejection with lower than conventional doses of immunosuppressive therapy [191,192,193,194]. However, to be effective, autologous MSCs should be administered before transplantation [189,195]. Indeed, since systemically infused MSCs tend to migrate to damaged tissues, such as those exposed to ischaemia/reperfusion injury [196,197], MSC infusion performed a few days after transplant led to intragraft MSC recruitment, subsequent graft inflammation, neutrophil infiltration, and C3 deposition, with no advantage for graft survival [189]. However, a pretransplant approach is not applicable to deceased donor transplantation since the surgical transplant procedure takes places a few hours after a donor kidney becomes available.

Casiraghi et al. recently investigated how to render MSCs a therapeutic option for recipients of organs from deceased donors as well. Based on evidence that human MSCs express C3aR and C5aR [198], and that the chemoattractant effects of anaphylatoxins could guide MSCs toward injured tissues [198,199], the authors evaluated the effect of antagonising complement receptors on MSCs given on day +2 post-transplant in preventing their recruitment into the graft and in prolonging graft survival [200]. Post-transplant MSC infusion combined with a short course of C3aR or C5aR antagonist, or the administration of MSCs pre-treated with C3aR and C5aR antagonists, prevented intragraft recruitment of MSCs [200]. Notably, antagonising C3aR or C5aR allowed MSCs to home the secondary lymphoid organs, and led to diminished C3 deposition and neutrophil recruitment, with a subsequent reduction in graft inflammation [200].

In conclusion, inhibiting the complement cascade not only might help prevent IRI and DGF, as well as alleviating both T cell- and antibody-mediated rejection (as discussed above) but could also be used to make the promising pro-tolerogenic MSC therapy feasible in the context of transplantation with organs from deceased donors.

## Figures and Tables

**Figure 1 cells-12-00791-f001:**
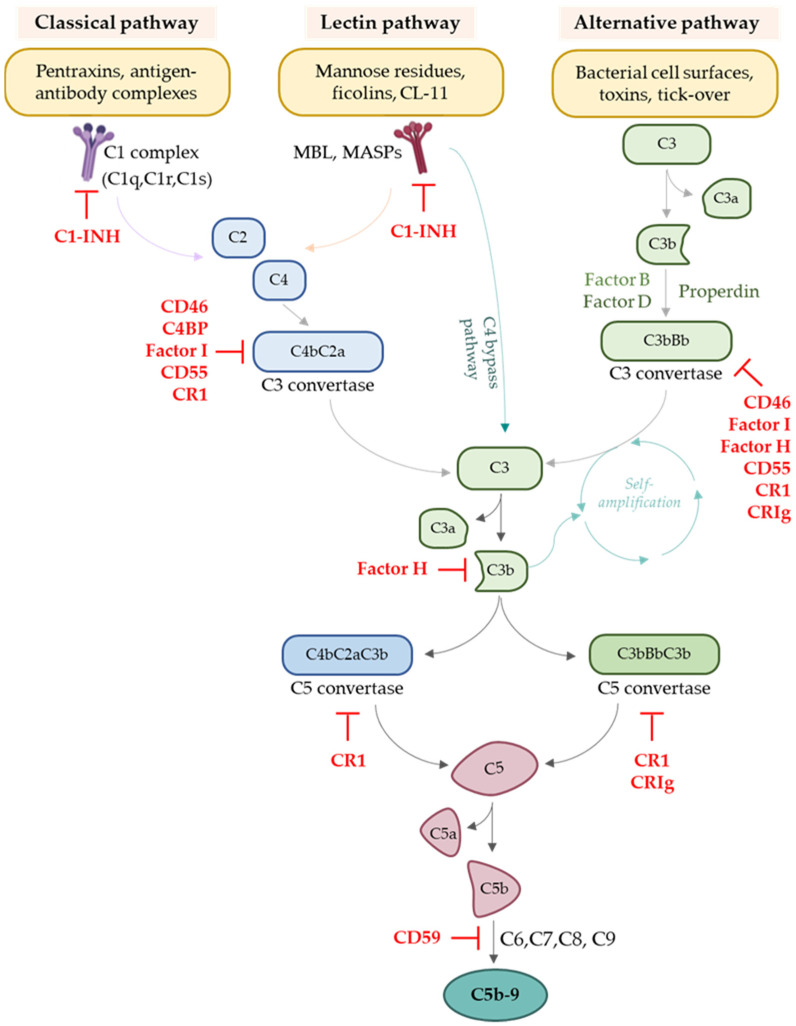
Schematic overview of the complement cascade and of its major ligands and regulators.

**Figure 2 cells-12-00791-f002:**
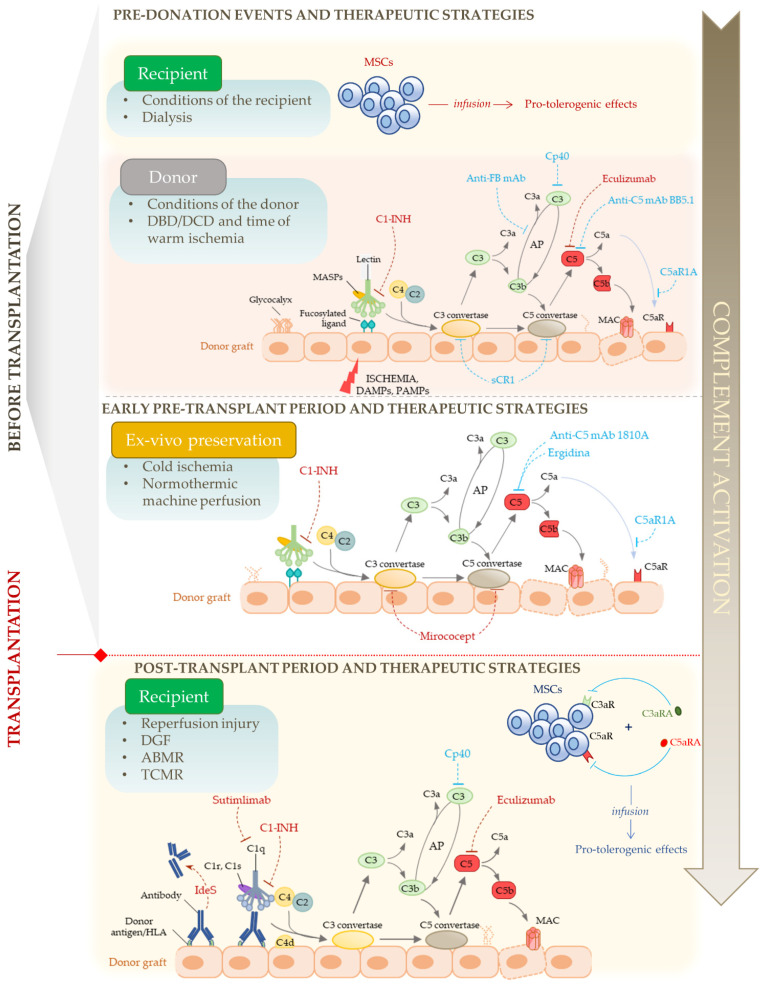
Schematic representation of the involvement of complement cascade activation in the sequential steps of kidney transplantation. Complement system might be activated before organ procurement, both in the recipient (e.g., for primary medical conditions and/or during dialysis sessions), and in the donor, whose pre-existing conditions significantly influence the graft quality and outcome. Moreover, the warm ischemia in the donor (during organ procurement and in DBD/DCD donation) and the cold ischemia during ex vivo preservation of the organ determine the release of DAMPs and increase the expression of fucosylated molecules by the ischemic injured tissue, triggering the activation of the lectin pathway, which is then amplified by the alternative pathway (AP). The ischemic conditions also disrupt the glycocalyx, with subsequent loss of complement regulators from endothelial cells. After transplantation, the kidney graft undergoes reperfusion injury with, in some cases, delayed graft function (DGF), and encounters the immune response of the recipient. Complement is involved both in T-cell mediated rejection (TCMR, not shown in the illustration) and in antibody-mediated rejection (ABMR). During ABMR, antibodies of the recipient bind to donor antigens (mainly donor HLA molecules) on the surface of the graft, and initiate the activation of the complement system through the classical pathway. In recipients, infusion of autologous mesenchymal stromal cells (MSCs) before or after (in this case, in combination with inhibitors of C3aR and of C5aR) transplantation recently emerged as a promising strategy aimed at dampening anti-donor immune response. The blue and red dashed lines indicate the complement inhibitors evaluated in pre-clinical models and in clinical trials, respectively.

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
