# Peer review of "The Complement System in Kidney Transplantation"

_cells, 2023, doi:10.3390/cells12050791_

Round 1
Reviewer 1 Report
The review presented by Santarsiero et al. describes in detail the role of the complement system at each stage of the kidney transplantation process. In particular, the authors highlight possible triggers of complement activation in kidney transplantation and therapeutic strategies that aim to inhibit the complement system to improve transplant success.
The manuscript is well presented but the following aspects should be revised by the authors:
- - The manuscript has superfluous statements such as: lines 40-46;
- - Novelty is limited since there are similar reviews recently published. Authors should underline the novel aspects of their paper (DOI: 10.1016/j.semnephrol.2022.01.006, DOI: 10.3389/fimmu.2019.02306);
- - The conclusions of the manuscript are too "strong”, it would be necessary to say that inhibition of the complement cascade not only MIGHT help prevent IRI and DGF...
Author Response
The review presented by Santarsiero et al. describes in detail the role of the complement system at each stage of the kidney transplantation process. In particular, the authors highlight possible triggers of complement activation in kidney transplantation and therapeutic strategies that aim to inhibit the complement system to improve transplant success.
The manuscript is well presented but the following aspects should be revised by the authors:
- The manuscript has superfluous statements such as: lines 40-46;
A: Following the Reviewer’s suggestion, we have modified the sentence making it less superfluous (lines 46-47).
- Novelty is limited since there are similar reviews recently published. Authors should underline the novel aspects of their paper (DOI: 10.1016/j.semnephrol.2022.01.006, DOI: 10.3389/fimmu.2019.02306).
A: We are aware that the subject of our review has been addressed by important authors and that similar reviews have been recently published. We thank the Reviewer for suggesting us two very recent reviews that we have quoted in the revised version of the review.
The activation of complement system plays a crucial role in every step of the transplantation process, from pre-donation events to the early pre-transplant and the post-transplant periods, as we summarized in Figure 2. The main aim of our review was to give a contribution in detailing and emphasizing which pathway and which complement components have a detrimental role in each transplantation phase, also underlining the results of pre-clinical and clinical studies that tested the efficacy of complement blockade as possible therapeutic targets. In doing that, we have highlighted that even the novel anti-ischemic approaches (e.g. the procedure performed by ex vivo normothermic machine perfusion) or the pro-tolerogenic cell therapies (e.g. the post-transplant infusion of MSCs in the case of deceased donor transplantation) need the concomitant blockade of complement activation to improve their effectiveness.
At the end of the introduction paragraph of the revised version, we have added a few sentences (highlighted in red) that explain these aspects of our review (page 1, lines 30-32 and 36-40).
- The conclusions of the manuscript are too "strong”, it would be necessary to say that inhibition of the complement cascade not only MIGHT help prevent IRI and DGF.
A: We have modified the conclusions making them less strong.
Reviewer 2 Report
This is a very complete and well written review about the role of the complement system in kidney transplantation.
I just have some minor concerns.
Line 71, the authors may include in this paragraph that Figure 1 shows the 3 pathways of complement activation.
Line 292, change wt to capital letters (WT).
Line 653, this sentence should be at the end of figure 2.
Author Response
This is a very complete and well written review about the role of the complement system in kidney transplantation.
A: We thank the Reviewer for the supportive comments.
I just have some minor concerns.
Line 71, the authors may include in this paragraph that Figure 1 shows the 3 pathways of complement activation.
A: Following the Reviewer’s suggestion, we have added a sentence (highlighted in red) explaining that Figure 1 schematically overviews the three pathways of complement activation (page 2, lines 73-74).
Line 292, change wt to capital letters (WT).
A: We have corrected.
Line 653, this sentence should be at the end of figure 2.
A: We have corrected.
Round 2
Reviewer 1 Report
the authors modified the manuscript according to my suggestions.